# Dopamine-Dependent Ketamine Modulation of Glutamatergic Synaptic Plasticity in the Prelimbic Cortex of Adult Rats Exposed to Acute Stress

**DOI:** 10.3390/ijms24108718

**Published:** 2023-05-13

**Authors:** Lia Forti, Elona Ndoj, Jessica Mingardi, Emanuele Secchi, Tiziana Bonifacino, Emanuele Schiavon, Giulia Carini, Luca La Via, Isabella Russo, Marco Milanese, Massimo Gennarelli, Giambattista Bonanno, Maurizio Popoli, Alessandro Barbon, Laura Musazzi

**Affiliations:** 1Department of Biotechnology and Life Sciences, Center for Neuroscience Research, University of Insubria, 21052 Busto Arsizio, Italy; lia.forti@uninsubria.it (L.F.); emanuele.secchi44@gmail.com (E.S.); eschiavon1979@gmail.com (E.S.); 2Department of Molecular and Translational Medicine, University of Brescia, 25123 Brescia, Italy; e.ndoj@unibs.it (E.N.); giulia.carini@unibs.it (G.C.); luca.lavia@unibs.it (L.L.V.); isabella.russo@unibs.it (I.R.); massimo.gennarelli@unibs.it (M.G.); alessandro.barbon@unibs.it (A.B.); 3School of Medicine and Surgery, University of Milano-Bicocca, 20900 Monza, Italy; jessica.mingardi@unimib.it; 4Unit of Pharmacology and Toxicology, Department of Pharmacy, University of Genoa, 16148 Genoa, Italy; tiziana.bonifacino@unige.it (T.B.); marco.milanese@unige.it (M.M.); giambattista.bonanno@unige.it (G.B.); 5Genetics Unit, IRCCS Istituto Centro S. Giovanni di Dio, Fatebenefratelli, 25125 Brescia, Italy; 6IRCCS Ospedale Policlinico San Martino, 16132 Genoa, Italy; 7Dipartimento di Scienze Farmaceutiche, Università Degli Studi di Milano, 20133 Milano, Italy; maurizio.popoli@unimi.it

**Keywords:** acute stress, synaptic plasticity, LTP, prefrontal cortex, ketamine, glutamate receptors

## Abstract

Traumatic stress is the main environmental risk factor for the development of psychiatric disorders. We have previously shown that acute footshock (FS) stress in male rats induces rapid and long-lasting functional and structural changes in the prefrontal cortex (PFC), which are partly reversed by acute subanesthetic ketamine. Here, we asked if acute FS may also induce any changes in glutamatergic synaptic plasticity in the PFC 24 h after stress exposure and whether ketamine administration 6 h after stress may have any effect. We found that the induction of long-term potentiation (LTP) in PFC slices of both control and FS animals is dependent on dopamine and that dopamine-dependent LTP is reduced by ketamine. We also found selective changes in ionotropic glutamate receptor subunit expression, phosphorylation, and localization at synaptic membranes induced by both acute stress and ketamine. Although more studies are needed to understand the effects of acute stress and ketamine on PFC glutamatergic plasticity, this first report suggests a restoring effect of acute ketamine, supporting the potential benefit of ketamine in limiting the impact of acute traumatic stress.

## 1. Introduction

Acute traumatic stress is a major risk factor for psychiatric disorders [1,2]. Indeed, trauma exposure not only can precipitate diseases classified in the Diagnostic and Statistical Manual of Mental Disorders (DSM) V as stress-related, such as posttraumatic stress disorder (PTSD), acute stress disorder, or reactive attachment disorder, but may also trigger other psychiatric disorders, including major depression and anxiety [2,3,4]. Although the stress response is physiological in nature, being aimed at coping with environmental changes, in stress-susceptible subjects, adaptive processes can turn into maladaptive ones, thus increasing the psychopathological risk [5]. Importantly, this risk increases with the severity of the traumatic event, due to mechanisms depleting coping reserves [1,6]. A typical example is provided by wars or environmental disasters such as earthquakes, which can exacerbate PTSD in over 40% of the exposed population.

In this context, animal models are crucial for studying the functional and molecular consequences of traumatic stress in the brain. Several brain areas have been implicated in the stress response, including the amygdala, hippocampus, and prefrontal cortex (PFC) [7,8]. Among them, the PFC has been involved in a variety of social, cognitive, and affective functions that are commonly disrupted in mental illness. Importantly, the PFC plays key roles in fear conditioning, retention of extinction, emotion regulation, and avoidance [9,10,11].

Over the last dozen years, our research group dissected the time-dependent functional and morphological alterations induced by acute unpredictable and unavoidable footshock (FS) stress in the PFC of male rats [1,12,13]. We reported that FS remarkably increases glutamate release and transmission in PFC soon after stress and up to 24 h later, together with inducing a simplification of apical dendrites in prelimbic PFC layers II–III pyramidal neurons, which lasts up to 14 days after stress. We have also recently found that in phenotypically vulnerable animals, carrier-mediated glutamate release from astroglia perisynaptic processes is increased as well, suggesting a long-term remodeling of tripartite glutamate synapses in the PFC of FS-stressed rats [14].

The non-competitive N-methyl-D-aspartate (NMDA) receptor antagonist ketamine is a fast-acting psychotropic drug, approved in the management of treatment-resistant depression and recently proposed as a promising therapeutic agent for other mood disorders as well, including PTSD (as recently reviewed in [15,16,17]). We have previously reported that ketamine facilitates fear memory extinction and reverses both the increase in glutamate transmission and dendritic remodeling in the PFC when administered 6 h after FS stress exposure [18].

Overall, our previous data show that acute FS can induce rapid and long-lasting behavioral, functional, and structural changes in the PFC of rats, while acute ketamine may reverse at least part of these alterations.

Previous evidence demonstrated that synaptic plasticity, the ability of synaptic connections between neurons to be weakened or strengthened, specifically in relation to long-term potentiation (LTP) and long-term depression (LTD), is severely influenced by traumatic stress exposure [7,19]. Acute stress was consistently reported to impair LTP in the hippocampus, as well as in the hippocampus–PFC and the amygdala–PFC projections [20,21,22], while hippocampal LTD is enhanced [7]. Less is known about stress-related changes in plasticity within the PFC [15,23].

In the present study, we asked whether acute subanesthetic ketamine administration 6 h after FS may have any impact on neuronal plasticity and expression or activation by phosphorylation of ionotropic glutamate receptor subunits in the PFC of rats.

## 2. Results

### 2.1. Dopamine-Dependent Modulation of Long-Term Potentiation in Prelimbic Cortex of Rats Subjected to Acute Footshock Stress and Ketamine Treatment

To assess the effects of acute FS stress and subanesthetic ketamine on synaptic plasticity in the prelimbic (PrL) medial PFC (mPFC), we used electrophysiological field recordings of brain slices from control animals, animals subjected to FS, and animals subjected to FS, injected with 10 mg/kg ketamine 6 h later, and sacrificed 24 h after the beginning of stress (Figure 1a). 

Because adult rats were used in the present study, we preliminarily confirmed previous data obtained in adolescent mice [24], showing that field potentials recorded in PrL layer 1 (L1) evoked by stimulation of nearby L1 afferents are mainly dependent on activation of glutamate synapses. Hence, they can be defined as field excitatory postsynaptic potentials (fEPSPs). Indeed, the field potentials were nearly completely blocked by the application of selective antagonists of α-amino-3-hydroxy-5-methyl-4-isoxazolepropionic acid (AMPA) and NMDA receptors (see Section 4 and Appendix A).

We then evaluated in the experimental groups long-term potentiation (LTP) of L1 fEPSPs, induced by high-frequency stimulation (HFS) of L1 afferents (Figure 1).

We first assessed LTP induction in slices bathed in normal artificial cerebrospinal fluid (aCSF) (Figure 1b,c).

We found that HFS produced a small negative long-term change in mean fEPSP peak (Figure 1b). Indeed, the relative fEPSP peak change 30 min after HFS (ΔfEPSP/fEPSP_0_) was −5.1 ± 1.6% in control animals and was not significantly affected by FS (−1.0 ± 3.6%) or ketamine after FS (5.9 ± 3.6%; Kruskal–Wallis test, *p* = 0.1433). However, while HSF induced negative mean fEPSP peak changes in all control animals, in most FS-stressed animals, especially when treated with ketamine, a small positive long-term change in mean fEPSP peak was observed (Figure 1c). Overall, these results confirm previous evidence collected in juvenile and adolescent animals showing that tetanic stimuli applied to L1/L2 do not evoke significant LTP of fEPSPs in prelimbic mPFC slices bathed in control saline [24,25].

Because previous studies showed that dopamine (DA) has a permissive role in LTP induction in mPFC slices of young animals [25,26], we evaluated whether the same also happened in adult rats, testing the effect on LTP of two different protocols of DA application to mPFC slices from unstressed control animals, compared to aCSF (Figure 2). Firstly, DA 50 μM was transiently added to aCSF for 10 min before HFS, using a protocol previously shown to facilitate LTP in the PFC of animals exposed to acute mild stress [26]. We found that DA 50 μM application induced a mild facilitation of LTP (Figure 2b), which did not reach significance compared to aCSF (Figure 2b). We then adopted a second protocol in which aCSF was supplemented with 3 μM DA starting at least 30 min before HFS, in the presence of picrotoxin (1 μM) to partially block GABAa receptor transmission [25] (Figure 2c). Using 3 μM DA yielded a significant LTP increase compared to aCSF (ΔfEPSP/fEPSP_0_, aCSF: −5.1 ± 1.6%; 50 μM DA: 12.7 ± 8.6%; 3 μM DA: 21.3 ± 5.6%; Kruskal–Wallis test *p* = 0.0159; Dunn’s multiple-comparisons test: 3 μM DA vs. aCSF, *p* = 0.0273) (Figure 2d). 

We thus selected the 3 μM DA protocol and studied the effect of FS and ketamine after FS on LTP induction in PFC slices in the presence of 3 μM DA. In this condition, HFS induced an easily detectable long-term change in fEPSP peak, FS did not significantly change LTP, and ketamine after FS slightly inhibited LTP (Figure 3a,b; ΔfEPSP/fEPSP_0_, CTR: 21.3 ± 5.6%; FS: 17.5 ± 3.2%; FS+KET: −2.1 ± 5.0%; Kruskal–Wallis test *p* = 0.0269; Dunn’s multiple-comparisons test: FS+KET vs. CNT, *p* = 0.050). 

Overall, our data show that 24 h after FS, there is no FS-induced change in LTP of L1 excitatory synapses in the prelimbic mPFC, while ketamine treatment after FS inhibits LTP.

### 2.2. Changes in Dopamine Release Induced by Acute Footshock Stress in the Prefrontal Cortex

Having confirmed that DA is required for the induction of LTP in the PFC of young-adult rats, we asked whether acute FS may alter DA release in the PFC, because previous evidence showed that stress increases DA release and transmission in mesolimbic dopaminergic pathways [27]. Here, we measured basal and depolarization-evoked dopamine release from isolated presynaptic terminals (synaptosomes) in superfusion of rats subjected to FS stress and sacrificed 6 h after the stress session (the same time selected in previous experiments for ketamine injection), compared to controls. We found no significant changes in basal DA release despite a trend of increase in stressed animals (Figure 4a), while depolarization-evoked DA release was remarkably increased in the PFC of FS-stressed rats compared to controls (Figure 4b; unpaired t-test *p* < 0.05). Unfortunately, due to the complexity of these experiments and to the limited number of animals available for experimentation, it was not possible to measure whether the increase in DA release was sustained up to 24 h after FS stress exposure or to verify whether ketamine injection was able to modulate the effects of stress. Nevertheless, albeit preliminary, these results show that one session of FS triggers a long-lasting activation of dopaminergic transmission within PFC.

### 2.3. Modulation of Ionotropic Glutamate Receptor Subunit Expression and Phosphorylation Levels Induced by Acute Footshock and Ketamine in The Prefrontal Cortex

To evaluate whether the changes in plasticity induced by FS and subanesthetic ketamine were accompanied by alterations of glutamate receptors in PFC, we measured protein expression and regulation by phosphorylation of major NMDA and AMPA receptor subunits. Western blot analyses were performed on total cell homogenate and synaptic membranes from the PFC of animals subjected to FS, injected with ketamine 6 h after, and sacrificed 24 h after stress exposure.

In PFC total homogenate, although the levels of the obligatory GluN1 NMDA subunit were not changed among the experimental groups (Figure 5a), GluN2A protein expression was increased after FS and ketamine treatment reversed the increase (Figure 5b; one-way ANOVA F(2,19) = 9.265, *p* = 0.0016; Tukey’s post hoc test: FS vs. CNT, *p* = 0.0158; FS+KET vs. FS, *p* = 0.0013); ketamine also reduced GluN2B levels compared to both control and FS-stressed groups (Figure 5c; one-Way ANOVA F(2,20) = 9.868, *p* = 0.0010; Tukey’s post hoc test: FS+KET vs. CNT, *p* = 0.0418; FS+KET vs. FS, *p* = 0.0008). No alteration was observed in the GluN2A/GluN2B ratio (Figure 5d).

Considering AMPA subunits, no alterations were observed for total GluA1 (Figure 6a) and its phosphorylation levels at Ser831 in the total homogenate (Figure 6b). Conversely, GluA1 phosphorylation levels at Ser845 were significantly reduced in all stressed animals with no main effect of ketamine (Figure 6c; one-Way ANOVA F(2,20) = 8.839, *p* = 0.0018; Tukey’s post hoc test: CNT vs. FS, *p* = 0.0377; CNT vs. FS+KET, *p* = 0.0014). As regards GluA2, its total expression was decreased in the FS+KET group compared to control (Figure 6d; one-Way ANOVA F(2,21) = 4.291, *p* = 0.0274; Tukey’s post hoc test: FS+KET vs. CNT, *p* = 0.028), while its phosphorylation at Ser880 was increased by FS and brought back to control levels by ketamine treatment (Figure 6e; one-Way ANOVA F(2,20) = 5.570, *p* = 0.0119; Tukey’s post hoc test: CNT vs. FS, *p* = 0.0206; FS vs. FS+KET, *p* = 0.0234).

A different pattern was observed at the synaptic level. In PFC synaptic membranes, no significant changes were observed in the GluN1 subunit (Figure 7a).

Moreover, although the levels of GluN2A and GluN2B were not altered (Figure 7b,c), the GluN2B/GluN2A ratio was significantly increased in animals subjected to FS compared to controls (Figure 7d; one-way ANOVA F(2,19) = 4.553, *p* = 0.0242; Tukey’s post hoc test: CNT vs. FS, *p* = 0.0268); the treatment with ketamine partly reversed this change (FS+KET vs. CNT, *p* = 0.8210). Regarding AMPA receptors, we found no significant changes in either GluA1 or GluA2 total expression or phosphorylation (Figure 8).

## 3. Discussion

The data collected here show that a single injection of subanesthetic ketamine 6 h after acute FS dampens DA-dependent LTP in PrL mPFC, while both FS and ketamine induce specific changes in ionotropic glutamate receptor subunit expression, phosphorylation, and localization at synaptic membranes.

While we have previously described the effects of acute FS on excitatory transmission in L2/3 pyramidal neurons, showing an enhancement in glutamate release and excitatory spontaneous activity up to 24 h after stress that was reversed by acute ketamine [18,28], possible changes in long-term plasticity (LTP induction) were never explored before.

Previous evidence showed that in slices from adolescent animals, presumably due to the lack of basal tone of endogenous regulators, synaptic potentiation is lost with standard LTP induction protocols using ≤100 Hz tetani. On the other hand, the external supply of a low DA tone in the presence of partially inhibited GABAa transmission was described as reversing the potentiation of excitatory pathways activated by stimulation of upper cortical layers (reviewed in [29]). In our work, we tested for the first time FS-stress-induced changes in LTP in the PrL mPFC of young-adult rats (7–9 weeks). In line with the results obtained in adolescent rats, while we did not observe any significant LTP induction by 100 Hz tetani in CSF, the application of low DA (3 μM) in the presence of a partial GABAa block recovered a significant LTP. Moreover, while FS stress did not affect LTP, potentiation was significantly inhibited by ketamine administered 6 h after FS.

In previous studies, the effects of mild acute stress (elevated platform) on LTP of different afferent pathways to PrL mPFC were tested both in vivo in adult rats and in vitro in adolescent rat slices. In vivo studies showed a decrease in LTP immediately after an elevated platform in both the basolateral amygdala–PrL cortex pathway [20] and in the ventral hippocampus–PrL cortex pathway, which was prevented by acute antidepressant injection [21]. Differently, in adolescent rat slices exposed to DA without GABAa blockers (thus lacking the control of inhibitory inputs), LTP of the L2-L5 PrL pathway was absent in control animals, unveiled 1 h after an elevated platform, and abolished by acute desipramine (but not subanesthetic ketamine) injection immediately after stress [26]. The inconsistency between in vivo and in vitro studies could originate not only from the limited circuitry preserved in slices but also from the different age of the animals and specific pathways analyzed.

Overall, these results and our previous findings suggest that changes in PrL mPFC LTP after stress may depend on the stressor type and intensity, time point of measurement after stress exposure, age of the animals, DA levels, control of inhibitory inputs, and specific pathways analyzed.

In this context, our results are consistent with previous literature demonstrating that DA dose-dependently enhances LTP in the PFC [25,26,29]. Importantly, DA transmission is remarkably affected by stress, which was consistently reported to rapidly increase mPFC DA levels [30,31]. In line with this, we found a remarkable increase in DA presynaptic release in the PFC of rats 6 h after stress. Although we did not have the opportunity to verify whether DA release was also altered 24 h after FS stress and if ketamine could exert any effect at this level, this is the first study reporting changes in DA release after FS directly measured with the technique of presynaptic terminals in superfusion. Most of the previous evidence, indeed, comes from microdialysis studies. However, it should be considered that the amount of neurotransmitters sampled by microdialysis is only indirectly correlated to presynaptic release, while the superfusion of synaptosomes allows evaluating presynaptic activity precisely and selectively, without any interference of reuptake and indirect effects [32]. Considering the key role of DA in modulating LTP in PrL mPFC, we recognize that measurements in slices, lacking an intact dopaminergic circuitry and with an artificial control of DA level, would limit the possibility to see changes in LTP induced in vivo by stress and drug treatments. For example, we could speculate that the increase in DA release induced by FS stress might alter LTP in vivo.

As for ketamine, intriguingly, previous evidence showed that acute ketamine administration in rodents is associated with a significant increase in DA levels in the cerebral cortex, striatum, and nucleus accumbens [33]. Nevertheless, in most of the studies, ketamine was administered to control and not stressed animals, and to the best of our knowledge, there is only one study evaluating changes in DA activity induced by ketamine administration in a stress model. However, it considered only the ventral tegmental area and the nucleus accumbens [34].

In this context, we have previously found that ketamine exerts a modulatory homeostatic effect on glutamatergic transmission, increasing or decreasing glutamate release depending on whether it is decreased (as in the case of chronic stress, [35]) or increased (as for acute stress, [18]), thus stabilizing glutamatergic transmission. Considering the interplay between glutamate and DA transmissions [36], we may speculate that ketamine could also exert a homeostatic effect on DA transmission, eventually normalizing the alterations induced by stress. More studies are needed to understand the role of DA in the modulation of PrL mPFC LTP under stress conditions and the impact of antidepressant ketamine at this level.

In the present study, we limited the analysis to the PFC. However, other brain areas are involved in the response to acute traumatic stress. For example, the hippocampus plays important roles in the contextual processing of fear, as well as in inhibitory avoidance [37]. It would thus be interesting to test whether ketamine can reverse the well-characterized impairment of hippocampal LTP induced by traumatic stress [22]. Intriguingly, previous studies reported that ketamine restores hippocampal LTP in animal models of depression [38,39], but, to the best of our knowledge, the effect of ketamine on hippocampal plasticity after acute traumatic stress has never been assessed before.

Synaptic plasticity is dependent on glutamate receptor composition at excitatory synapses [40], and previous studies demonstrated that psychotropic effects are associated with changes in ionotropic glutamate receptor expression and phosphorylation [41,42,43]. Thus, we analyzed the expression and phosphorylation patterns of the main NMDA and AMPA receptor subunits in both total extracts and synaptic fractions of PFC.

As for NMDA receptors, in PFC total homogenate, we found increased GluN2A subunit protein expression in FS-stressed animals, which was completely reversed to control levels by acute ketamine; ketamine also reduced GluN2B expression. Differently, in synaptic membranes, the GluN2A/GluN2B ratio was increased in the FS group, and ketamine treatment partly reversed this upregulation.

NMDA receptors are heteromeric receptors composed of two constitutive GluN1 subunits and two regulatory subunits, predominantly GluN2A or GluN2B [40]. The latter play key roles in determining the functional properties of NMDA receptors: GluN2A-containing receptors have higher open probability and deactivate and desensitize faster than GluN2B-containing NMDA receptors [44,45,46]. Our data show an overall reduction in the expression of both GluN2A and GluN2B induced by ketamine, whereas at synaptic membranes, the increase in GluN2A/GluN2B ratio induced by stress suggests an enrichment at synapses of NMDA receptors with fast kinetics, with possible consequences for synaptic plasticity [40]. On the other hand, acute ketamine partly reversed this alteration, in line with a normalization of synaptic function and possibly related with the observed reduction in LTP.

Considering AMPA receptors, interestingly, we observed more changes in the total extracts than at the synaptic level. Indeed, we found a reduction in GluA1 phosphorylation at Ser845 in all stressed animals, decreased expression of GluA2 in stressed animals treated with ketamine, increased GluA2 phosphorylation at Ser880 in FS animals in the total homogenate, and no significant changes at synaptic membranes. Phosphorylation of GluA1 at Ser845 increases opening probability, as well as peak amplitude, of AMPA receptor currents and is implicated in AMPA receptor trafficking to the synaptic sites [47,48], while phosphorylation of GluA2 at Ser880 promotes AMPA receptor internalization [48,49]. Thus, our data suggest that FS stress may induce a general dampening of AMPA receptor function, while ketamine only partly reverses this effect. Nevertheless, because these changes were found in the homogenate and not at the synaptic level, we speculate that they involve extra-synaptic and intracellular pool receptors, thus suggesting that changes in synaptic AMPA receptors could occur at earlier time points after acute stress exposure, while 24 h after stress, an equilibrium has been reached with no major changes in AMPA receptors at synapses.

## 4. Materials and Methods

### 4.1. Animals

All experimental procedures involving animals were performed in accordance with the European Community Council Directive 2010/63/UE and were approved by the Italian legislation on animal experimentation (Decreto Legislativo 26/2014, animal experimentation licenses 521/2015-PR, 505/2017-PR, 140/2014-B—DGSAF24898). Adult Sprague–Dawley male rats were used (350–450 g, Charles River, Calco, Italy). All the animals were housed 2 per cage and maintained on a 12/12 h light/dark schedule (lights on at 7:00 am), in a temperature-controlled facility with free access to food and water.

Rats were randomly assigned to 3 groups: CNT (naïve animals injected with vehicle), FS (animals undergoing acute FS stress protocol and receiving vehicle injection 6 h after the beginning of the stress session), and FS+KET (animals undergoing acute FS stress protocol and receiving 10 mg/kg ketamine injection 6 h after the beginning of the stress session). All the animals were sacrificed by beheading 24 h after the beginning of FS, and PFC was rapidly dissected on ice.

### 4.2. Drug Treatment

Racemic ketamine (MSD Animal Health, Milan, Italy), 10 mg/kg, i.p., in saline 0.9% [18,35].

### 4.3. Footshock Stress

Rats were subjected to a single session of acute inescapable FS stress as previously reported [18,50,51,52]: intermittent shocks (0.8 mA) for 40 min (20 min total of actual shock with random intershock length between 2 s and 8 s). The FS box was connected to a scrambler controller (LE 100-26, Panlab, Barcelona, Spain) that delivered intermittent shocks to the metallic floor. Control animals were left undisturbed in their home cages.

### 4.4. Brain Slices Preparation

Coronal brain slices (400 μm thickness) containing the prelimbic (PrL) medial PFC (bregma 3.0–4.7 mm) were cut in ice-cold solution and maintained at air–saline interface at 34 C° for 30 min. Then, they were cooled down at room temperature until use 1–6 h from the end of the slice cutting. The cutting and maintenance saline contained the following (mM): NaCl (83), KCl (2.5), NaH_2_PO_4_ (1.25), NaHCO_3_ (21), glucose (25), sucrose (72), Na-ascorbic acid (0.45), CaCl_2_ (1), and MgCl_2_ (4), constantly carboxygenated with 95%O_2_, 5%CO_2_.

### 4.5. Field Recordings of Excitatory Postsynaptic Potentials (fEPSPs)

For recordings, slices were transferred to a submerged-style chamber and perfused (2 mL/min) with aCSF (33 ± 1 °C) containing the following (mM): NaCl (125), KCl (2.5), NaH_2_PO_4_ (1.25), NaHCO_3_ (26), D-glucose (10), CaCl_2_ (1.5), and MgCl_2_ (1).

Slices were visualized with differential interference contrast optics and infrared illumination. The PrL cortex was identified in accordance with the Paxinos and Watson rat brain atlas [53]. The stimulating and recording electrodes were positioned in the inner third of the PrL cortical layer 1 (L1), at 100 μm from each other, along the transverse axis [24]. Current clamp data were acquired with a Multiclamp 700B, low-pass-filtered at 10 KHz and sampled at 50 KHz. Constant current stimuli (0.2 ms pulse duration) were delivered with a tungsten bipolar concentric electrode (50 μm tip), and evoked field potentials were recorded with an aCSF-filled glass pipette (~1–2 MΩ tip resistance). Test stimuli were delivered every 20 or 30 s (0.2 ms), as specified. The stimulus intensity was the current providing ~60% of the maximal response in the range 20–100 μA (generally 50–60 μA). Responses included a presynaptic volley (PV; ~0.8 ms from stimulus) and a postsynaptic potential (PSP; 2.5–3.5 ms from stimulus; Figure 1a and Figure 3a, insets). The excitatory nature of the negative-going PSP peak was tested by applying the AMPA receptor antagonist NBQX (2 μM; Abcam, Cambridge, UK) and the NMDA receptor antagonist D-(-)-2-Amino-5-phosphonopentanoic acid (D-APV) (50 μM; Abcam) (Appendix A) [24].

After obtaining a stable basal response for at least 10′, a HFS protocol was used to induce LTP, consisting of 4 trains at 100 Hz (1 s train duration, 0.2 ms stimuli, 20 s inter-train interval). In experiments using 3 μM DA (Sigma-Aldrich, Milano, Italy), the number of trains was increased to 5. In the latter case, DA perfusion started at least 30 min before acquisition of baseline fEPSPs and continued all along, in the continuous presence of the GABAa receptor blocker picrotoxin (1 μM; Tocris, Bristol, UK). DA was dissolved in aCSF before use, supplemented with 20 μM ascorbic acid to prevent oxidation.

### 4.6. Synaptosomes Preparation and Release Experiments

Isolated nerve terminals (synaptosomes) from frozen rat PFC were prepared according to [54]. Briefly, the tissue was homogenized in 40 volumes of 0.32 M sucrose, buffered to pH 7.4 with phosphate buffer (final concentration 0.01 M). The homogenate was centrifuged at 1000×g for 5 min, to remove nuclei and cellular debris, and crude synaptosomes were isolated from the supernatant by centrifugation at 12,000×g for 20 min. The synaptosomal pellet was then resuspended in a physiological medium with the following composition (mM): NaCl, 140; KCl, 3; MgSO_4_, 1.2; NaH_2_PO_4_, 1.2; NaHCO_3_, 5; CaCl_2_, 1.2; 4-(2-hydroxyethyl)-1-piperazineethanesulfonic acid (HEPES), 10; and glucose, 10; with pH 7.4.

Synaptosomes were incubated at 37 °C for 15 min in the presence of 0,1 µM [^3^H] dopamine ([^3^H] DA; 0.5 Ci/mmol; Perkin Elmer Boston, MA, USA), 0.1 µM 6-nitroquipazine, and 0.1 µM reboxetine, to avoid false labeling of serotonergic and noradrenergic nerve terminals. Furthermore, 1 mM ascorbic acid and 10 µM pargyline were also added to counteract oxidative processes of the [^3^H]DA tracer [55]. Aliquots were distributed on microporous filters placed at the bottom of a set of 24 parallel superfusion chambers maintained at 37 °C (Superfusion System, Ugo Basile, Comerio, Varese, Italy) [18], started with physiological medium at a rate of 0.5 mL/min, and continued for 48 min. After 39 min of superfusion to equilibrate the system, two 3 min samples (t = 36–39 and 45–48 min; basal release) and one 6 min sample (t = 39–45 min; stimulus-evoked release) were collected. A 90 s stimulation pulse of 20 mM KCl, substituting for an equimolar concentration of NaCl, was applied at t = 39 min. Samples and superfused synaptosomes were measured for radioactivity. Tritium released in each sample was calculated as a percentage of the total synaptosomal tritium content at the beginning of the respective collection period (fractional rate × 100). The stimulus-evoked overflow was estimated by subtracting the transmitter content in the two 3 min fractions, representing the basal release, from the 6 min fraction collected during and after the stimulation pulse.

### 4.7. Western Blotting

Protein quantification was carried out by bicinchoninic acid (BCA) concentration assay (Sigma-Aldrich). Western blot analysis was carried out as previously described by incubating nitrocellulose membranes containing electrophoresed and blotted proteins from either homogenate or presynaptic membrane [43,56]. Primary antibodies used were GluN1 (1:500, cod. AB9864, Millipore, Milano, Italy), GluN2A (1:500, cod. AB1555P, Millipore), GluN2B (1:500, cod. 454,582, Millipore), GluA1 (1:200, cod. AGC-004, Alomone Labs, Jerusalem, Israel), GluA2 (1:2500, cod. AGC-005, Alomone Labs), GluA1-pSer831 (1:1000, cod. Ab109464, Abcam), GluA1-pSer845 (1:1000, cod. ab3901, Abcam), and GluA2-pSer880 (1:000, cod. Ab52180, Abcam). Antibodies against α-Tubulin (1:20000, cod. ab7291, Abcam) or ꞵ-Actin (1:2000, cod. AB-81599, Immunological Science, Roma, Italy) were used as internal controls. Membranes were incubated with fluorophore-conjugated secondary antibodies (1:2000, IRDye 800CW goat anti-rabbit IgG 926-32211 or IRDye 680RD goat anti-mouse IgG 926-68020, LI-COR, Bad Homburg, Germany). Signals were detected using Odyssey infrared imaging system (LI-COR Biosciences) and quantified with Image Studio software (version 5.2, LI-COR Biosciences). Data are presented as optical density ratios of investigated protein bands normalized to α-Tubulin or ꞵ-Actin bands in the same line and are expressed as percentage of controls.

### 4.8. Analysis and Statistics

Normal distribution was verified using Kolmogorov–Smirnov test.

fEPSP and PV negative-going peaks were measured with respect to the voltage maximum at the end of PV using Clampfit (Molecular Devices). For quantification of LTP, fEPSP peaks were averaged in the 25–30′ time window after HFS and reported as relative change (ΔfEPSP/fEPSP_0_) with respect to the basal mean peak (fEPSP_0_), averaged in the 10′ before HFS. A few experiments with PV changes larger than 25% were not analyzed. The relative changes in fEPSP and PV were not correlated in the analyzed dataset (Spearman r: *p* = 0.124). Individual data points used for statistical comparison of experimental groups are “per animal” data obtained by averaging over slices from a single animal. Statistical comparison of ΔfEPSP/fEPSP_0_ between groups was performed using the Kruskal–Wallis test corrected for multiple comparisons, because in each comparison, at least one group did not pass the normality or log-normality test.

DA release experiments were analyzed using unpaired Student’s t-test, while for Western blotting, one-way analysis of variance (ANOVA) followed by Tukey’s post hoc test was applied.

Statistical comparisons were made with GraphPad Prism 9.4 (GraphPad Software). All data in the text are provided as mean ± SEM. Threshold for statistical significance was set to 0.05.

## 5. Conclusions

Overall, despite its limits, this is the first study analyzing changes in DA-dependent plasticity and glutamate receptor expression and phosphorylation in the PFC of adult male rats exposed to acute FS stress and ketamine treatment. Although the use of brain slices in functional studies prevented the physiological dopaminergic input, which could be modulated in stress conditions compared to control, thus in turn potentially affecting LTP in vivo, we found that the administration of ketamine 6 h after FS reduces DA-dependent plasticity. On the other hand, our molecular study demonstrated an enrichment in GluN2A-containing NMDA receptors at synapses 24 h after stress, suggesting altered plasticity, while ketamine reversed this change. Altogether, our results are in line with a modulatory homeostatic effect of ketamine on glutamatergic transmission, as previously suggested [18].

More studies are needed to understand how the exposure to a traumatic event may alter synaptic plasticity in the PFC. In particular, because previous evidence showed sex differences in conditioned fear and fear extinction [57], an important issue would be to test whether the effects of stress and ketamine are similar or different in the two sexes.

Nevertheless, this first report is in line with the field of research supporting the use of ketamine treatment early after trauma exposure to limit the long-lasting consequences of acute stress.

## Figures and Tables

**Figure 1 ijms-24-08718-f001:**
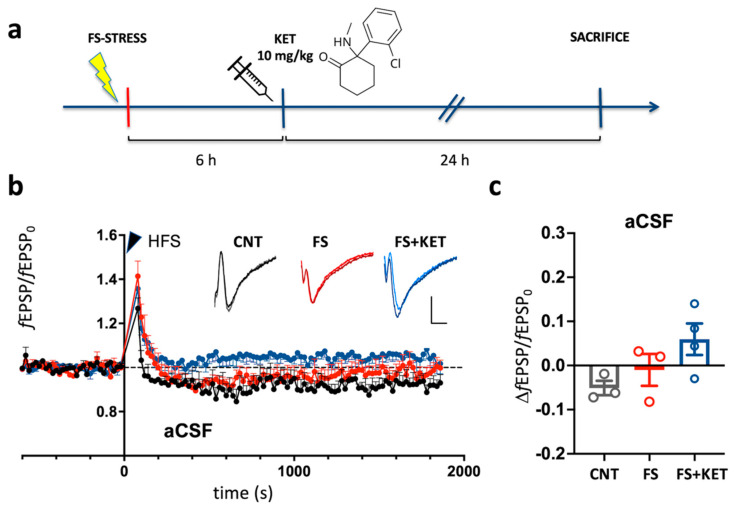
**Effects of acute footshock stress and ketamine on LTP in in adult prelimbic cortex slices bathed in aCSF.** (**a**) Timeline. Inset: molecular structure of ketamine. (**b**) Mean time course of fEPSP peak, normalized to the average peak during the 10′ pre-tetanus baseline period (fEPSP_0_,). fEPSPs evoked every 20 s. A tetanic stimulus (HFS: 100 stimuli at 100 Hz, 4 times every 20 s) was given at time = 0 (arrowhead). Black: control animals (CNT, 3 slices from 3 animals); red: FS-stressed animals (FS; 5 slices from 3 animals); blue: FS-stress + ketamine group (FS + KET, 6 slices from 4 animals). The means are obtained by averaging over mean time courses for each animal. Insets: representative fEPSPs (average of 5 responses) from individual slices from the CNT (left), FS (middle), and FS + KET (right) groups. For each slice, the fEPSP in the baseline period immediately preceding HFS (light color trace) is superimposed on the response 30′ after HFS (dark color trace). Calibration: 3 ms, 0.5 mV. (**c**) Relative fEPSP peak long-term change after HFS 30 min after HFS. Graph plots data for individual animals (dots) and their average (bars). Error bars: standard error of the mean (SEM).

**Figure 2 ijms-24-08718-f002:**
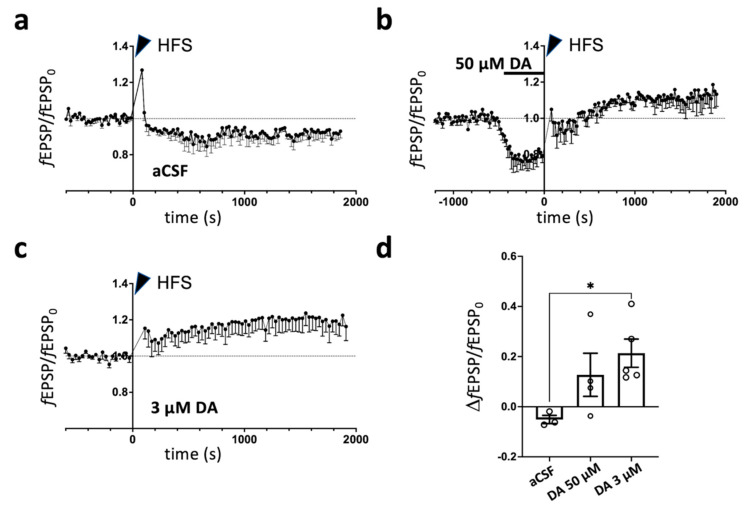
**Dopamine-dependent long-term plasticity of L1 fEPSPs in adult prelimbic cortex slices.** (**a**–**c**) Time course of fEPSP peak normalized to baseline (fEPSP_0_) for slices maintained in aCSF. A tetanic stimulus (HFS) was given at time = 0 (arrowheads). (**a**) Results for 3 slices from 3 animals maintained in aCSF. (**b**) Results for 6 slices from 4 animals bathed in aCSF with transient addition of 50 μM DA for 10 min before HFS. (**c**) Results for 6 slices from 5 animals maintained in aCSF supplemented with 3 μM DA and 1 μM picrotoxin. (**d**) Summary of relative fEPSP peak long-term changes 30 min after HFS for experiments in (**a–c**). * *p* < 0.05.

**Figure 3 ijms-24-08718-f003:**
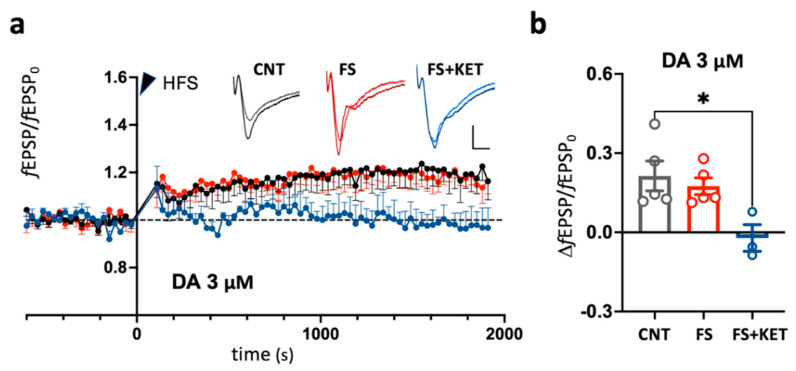
**Effects of footshock stress and ketamine on LTP in adult prelimbic cortex slices bathed in aCSF supplemented with 3 μM DA and 1 μM picrotoxin.** (**a**) Mean time course of fEPSP peak, normalized to baseline. fEPSPs evoked every 30 s. A tetanic stimulus (HFS; 100 stimuli at 100 Hz, 5 times every 20 s) was given at time = 0 (arrowhead). The mean is obtained by averaging mean time courses for each animal. Black: CNT (6 slices, from 5 animals); red: FS (8 slices from 5 animals); blue: FS+KET (5 slices from 3 animals). Insets: representative fEPSPs (average of 5 responses) from individual slices from the CTR (left), FS (middle), and FS + KET (right) groups. For each slice, the fEPSP in the baseline period immediately preceding HFS (light color traces) is superimposed on the response 30′ after HFS (dark color traces). Calibration: 3 ms, 0.5 mV. (**b**) Relative fEPSP peak long-term change after HFS. Data for individual animals (dots) and their average (bars). Error bars: SEM. Dunn’s multiple-comparisons test, * *p* < 0.05.

**Figure 4 ijms-24-08718-f004:**
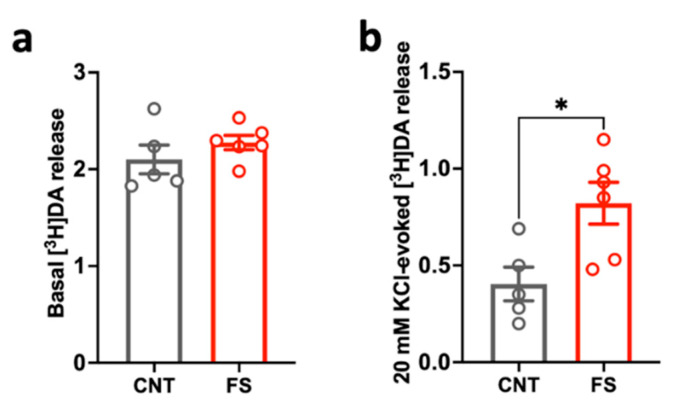
**Dopamine release from PFC synaptosomes of control and footshock-stressed rats.** (**a**) Basal [^3^H]DA release and (**b**) 20 mM KCl-evoked [^3^H]DA release measured in PFC synaptosomes in superfusion of CNT and FS rats sacrificed 6 h after stress exposure. Data are expressed as means ± SEM as basal efflux (fractional rate × 100) (**a**) and percent overflow (**b**). Unpaired Student’s t-test, * *p* < 0.05; N = 5 (CNT) and 6 (FS).

**Figure 5 ijms-24-08718-f005:**
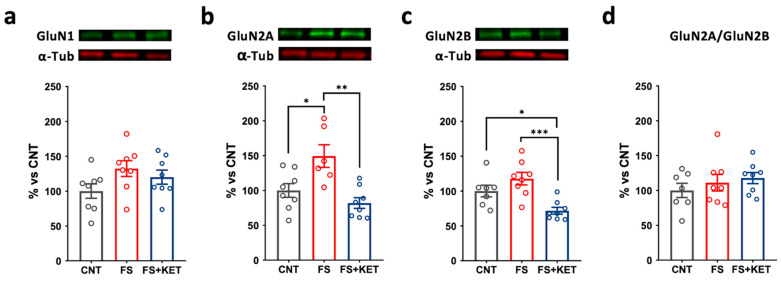
**Protein expression levels of NMDA receptor subunits in PFC total homogenate.** (**a**) GluN1, (**b**) GluN2A, (**c**) GluN2B, and (**d**) GluN2A/GluN2B ratio protein expression levels in PFC total homogenate from CNT, FS, and FS + KET rats. Data are expressed as percentage of controls and shown as means ± SEM. N = 8. Tukey’s post hoc test, * *p* < 0.05, ** *p* < 0.01, *** *p* < 0.001.

**Figure 6 ijms-24-08718-f006:**
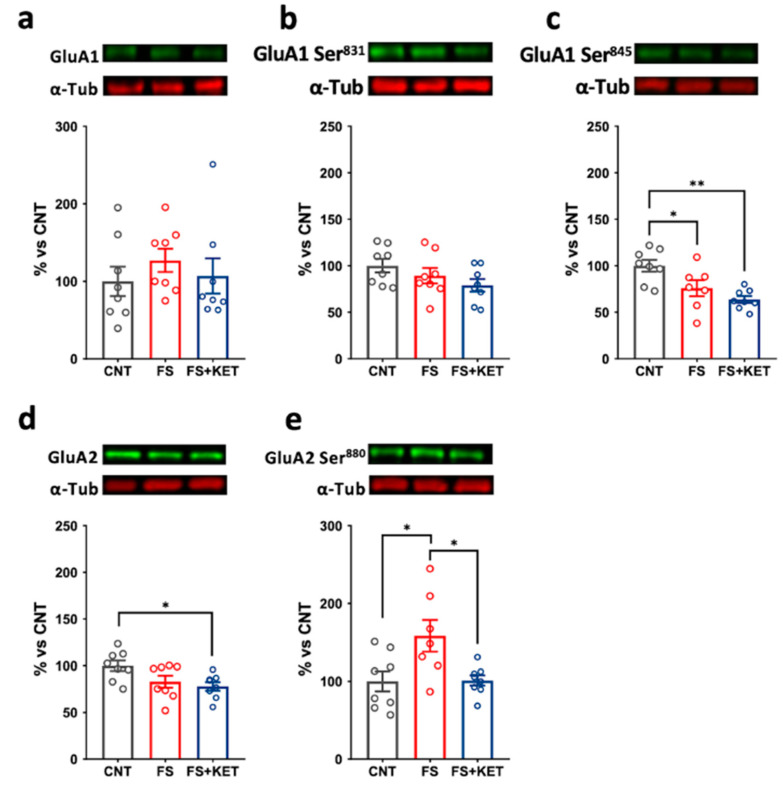
**Protein expression levels and phosphorylation of AMPA receptor subunits in PFC total homogenate.** (**a**) Total GluA1, (**b**) GluA1 phospho-Ser831, (**c**) GluA1 phospho-Ser845, (**d**) total GluA2, and (**e**) GluA2 phospho-Ser880 protein expression levels in PFC total homogenate from CNT, FS, and FS + KET rats. Data are expressed as percentage of controls and shown as means ± SEM. N = 8; Tukey’s post hoc test, * *p* < 0.05, ** *p* < 0.01.

**Figure 7 ijms-24-08718-f007:**
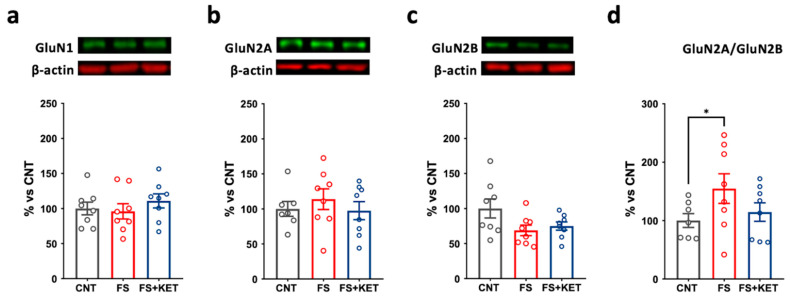
**Protein expression levels of NMDA receptor subunits in PFC synaptic membranes.** (**a**) GluN1, (**b**) GluN2A, (**c**) GluN2B, and (**d**) GluN2A/GluN2B ratio protein expression levels in PFC synaptic membranes from CNT, FS, and FS + KET rats. Data are expressed as percentage of controls and shown as means ± SEM. N = 8. Tukey’s post hoc test, * *p* < 0.05.

**Figure 8 ijms-24-08718-f008:**
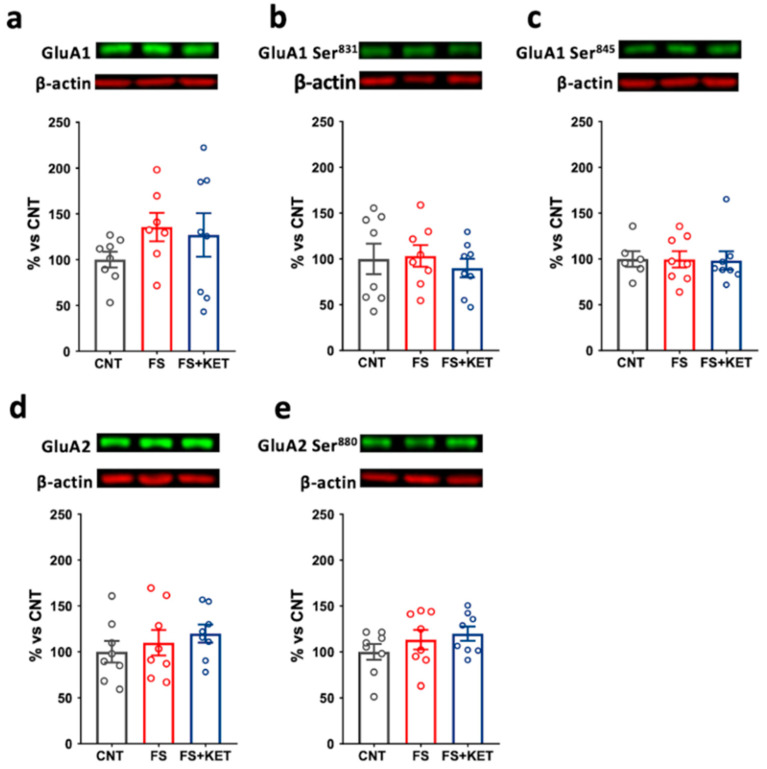
**Protein expression levels and phosphorylation of AMPA receptor subunits in PFC synaptic membranes.** (**a**) Total GluA1, (**b**) GluA1 phospho-Ser831, (**c**) GluA1 phospho-Ser845, (**d**) total GluA2, and (**e**) GluA2 phospho-Ser880 protein expression levels in PFC synaptic membrane from CNT, FS, and FS + KET rats. Data are expressed as percentage of controls and shown as means ± SEM. N = 8; Tukey’s post hoc test.

## Data Availability

Not applicable.

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
