# Peer review of "Dopamine-Dependent Ketamine Modulation of Glutamatergic Synaptic Plasticity in the Prelimbic Cortex of Adult Rats Exposed to Acute Stress"

_ijms, 2023, doi:10.3390/ijms24108718_

Round 1

Reviewer 1 Report

Psychopharmacology of both clinically applied antidepressants (SSRI and SNRI) and some  new promising medications such as NMDAR modulators in the context of anxiety, acute stress-related and affective processes  is currently an important topic in the contemporary psychiatry and neuroscience. Pharmacological effects of the SSRI in patients suffering from depression and anxiety disorders are usually detected after about four weeks of drug administration. However, some novel drugs e.g. ketamine may act starting even at the first dose, by changing the way individuals process affective information. Moreover, while SSRIs pharmacological response generally takes weeks before it is manifested clinically, an accumulating number of studies have examined the ability to predict the outcome of antidepressant therapy. There was also reported that there are some observable more or less subtle neurochemical changes that appear within hours of administering a single dose, suggesting that differential psychopharmacological effects can be present following acute rather than long-term SSRIs administration before detectable behavioural changes occur. On the other hand ketamine, as the NMDAR ionic channel blocker is considered as a fast-acting agent, approved in the treatment of depression and mood disorders and PTSD.

This very interesting electrophysiological and neurochemical study by Forti and colleagues shows for the first time that dopamine-dependent LTP in the rat prefrontal cortex slices is reduced by ketamine both in control and stressed animals. Furthermore, Authors revealed that ketamine as well as acute stress exerts several molecular changes, especially in the expression NMDARs and AMPARs subunits,  their phosphorylation and synaptic distribution. In my opinion, the Author’s conlusion suggesting that the stress-dependent enrichment of GluN2A-containing NMDARs at synapses may be restored by ketamine administration should definitely be highlighted.

Experimental design is, in general very well considered; all pharmacological procedures including drug administration, acute stress model, brain tissue preparation, fIPSP recording  and especially synaptosome isolation and Western blotting were kept the high standard.  Appropriate statistical methods were applied. The study is well documented and manuscript is sufficiently informative for the readership. All graphs, tables and WB images are clear an easy to follow. To sum up, this article may be considered as distinctly valuable contribution to the field of neuropsychopharmacology.

However, I have some minor suggestions for the Authors:

1. What is the justification for the ketamine dose 10mg/kg applied in the experiment? Have you got some references supporting this?

2. The study is focused on PFC, perhaps the hippocampus should also be examined using the same experimental design? It is a future direction, of course. What do Authors think about it?

3. Some changes in the NMDARs subunits expression were found in the rat hippocampus after long term treatment with several antipsychotics (Krzystanek et al. 2015, Pharmacological Reports). Although, its a different group of medications, this piece of information can potentially be worth mentioning in Discussion.

4. The structure of ketamine is rather known, however it is very unique and somehow peculiar from the chemical viewpoint, so I strongly recommend to add a small figure depicting its molecule (e.g. it can be iserted close to the graph). This may be attractive for the readers.

Author Response

Psychopharmacology of both clinically applied antidepressants (SSRI and SNRI) and some new promising medications such as NMDAR modulators in the context of anxiety, acute stress-related and affective processes is currently an important topic in the contemporary psychiatry and neuroscience. Pharmacological effects of the SSRI in patients suffering from depression and anxiety disorders are usually detected after about four weeks of drug administration. However, some novel drugs e.g. ketamine may act starting even at the first dose, by changing the way individuals process affective information. Moreover, while SSRIs pharmacological response generally takes weeks before it is manifested clinically, an accumulating number of studies have examined the ability to predict the outcome of antidepressant therapy. There was also reported that there are some observable more or less subtle neurochemical changes that appear within hours of administering a single dose, suggesting that differential psychopharmacological effects can be present following acute rather than long-term SSRIs administration before detectable behavioural changes occur. On the other hand ketamine, as the NMDAR ionic channel blocker is considered as a fast-acting agent, approved in the treatment of depression and mood disorders and PTSD.

This very interesting electrophysiological and neurochemical study by Forti and colleagues shows for the first time that dopamine-dependent LTP in the rat prefrontal cortex slices is reduced by ketamine both in control and stressed animals. Furthermore, Authors revealed that ketamine as well as acute stress exerts several molecular changes, especially in the expression NMDARs and AMPARs subunits,  their phosphorylation and synaptic distribution. In my opinion, the Author’s conlusion suggesting that the stress-dependent enrichment of GluN2A-containing NMDARs at synapses may be restored by ketamine administration should definitely be highlighted.

Experimental design is, in general very well considered; all pharmacological procedures including drug administration, acute stress model, brain tissue preparation, fIPSP recording  and especially synaptosome isolation and Western blotting were kept the high standard.  Appropriate statistical methods were applied. The study is well documented and manuscript is sufficiently informative for the readership. All graphs, tables and WB images are clear an easy to follow. To sum up, this article may be considered as distinctly valuable contribution to the field of neuropsychopharmacology.

We thank the reviewer for appreciating our work. As suggested, we have highlighted the restoring effect of ketamine on GluN2A expression (p. 14).

However, I have some minor suggestions for the Authors:

  1. What is the justification for the ketamine dose 10mg/kg applied in the experiment? Have you got some references supporting this?

The subanesthetic dose 10 mg/kg is the most widely used dose for racemic ketamine in preclinical studies on animal models of depression (Li et al., Science. 2010;329(5994):959-64. doi: 10.1126/science.1190287; Sala et al., Front Pharmacol. 2022;13:759626. doi: 10.3389/fphar.2022.759626. Tornese et al., Neurobiol Stress. 2019;10:100160. doi: 10.1016/j.ynstr.2019.100160). Appropriate references have been added to the main text (p. 12).

  1. The study is focused on PFC, perhaps the hippocampus should also be examined using the same experimental design? It is a future direction, of course. What do Authors think about it?

We agree with the referee that the study of the effect in the hippocampus would be of great interest. Indeed, the hippocampus plays important roles in the contextual processing of fear and inhibitory avoidance, besides its roles in memory and control of emotional response (Izquierdo et al., Physiol Rev. 2016;96(2):695-750. doi: 10.1152/physrev.00018.2015). In particular, it would be interesting to test whether ketamine can rescue the well-characterized impairments of LTP induced by traumatic stress (Ryan et al., Hippocampus. 2010 Jun;20(6):758-67. doi: 10.1002/hipo.20677). Intriguingly, previous studies reported that ketamine restores hippocampal LTP in animal models of depression (Aleksandrova et al., Mol Brain. 2020;13(1):92. doi: 10.1186/s13041-020-00627-z; Yang et al., Front Behav Neurosci. 2018;12:229. doi: 10.3389/fnbeh.2018.00229) but, to the best of our knowledge, the effect of ketamine on hippocampal plasticity after acute traumatic stress has never been assessed before. A comment about possible effects on the hippocampus has been added to the discussion (p.10-11). 

  1. Some changes in the NMDARs subunits expression were found in the rat hippocampus after long term treatment with several antipsychotics (Krzystanek et al. 2015, Pharmacological Reports). Although, its a different group of medications, this piece of information can potentially be worth mentioning in Discussion.

As suggested, in the revised text we cited a few papers showing changes of glutamate ionotropic receptors induced by psychotropic drugs, including the study mentioned by the referee (p. 11).

  1. The structure of ketamine is rather known, however it is very unique and somehow peculiar from the chemical viewpoint, so I strongly recommend to add a small figure depicting its molecule (e.g. it can be iserted close to the graph). This may be attractive for the readers.

As suggested by the referee, we added the molecular structure of ketamine to Figure 1.

Reviewer 2 Report

I have reviewed the manuscript “Dopamine-dependent ketamine modulation of glutamatergic 2 synaptic plasticity in the prelimbic cortex of adult rats exposed 3 to acute stress” by Lia Forti and coworkers. The authors study whether acute subanesthetic ketamine administration 6 h after receiving an acute stress (footshock) has any impact on neuronal plasticity (LTP) and expression or activation by phosphorylation of ionotropic glutamate receptors subunits in the pre frontal cortex of male rats.

I find the manuscript appropriate for publication in the International Journal of Molecular Sciences. However, I do have some suggestions and comments that I think will help the authors to improve the clarity for the reader of this article.

I suggest the authors to add more information concerning why they have chosen to test two different DA protocols (lines 127-133) in the results section. Although information is provided in the discussion section of why the authors firstly used 3 uM DA (lines 248-258) providing this information earlier in the manuscript will help the reader to better understand why those two experimental procedures were performed.

Why have the authors performed their experiments only on male rats? At least a comment for this preference should be stated. Acute traumatic stress can precipitate several psychiatric disorders and gender should be taken into account.

Although most abbreviatures are defined in the methodology section, it would help the reader to acknowledge them the first time of appearance in the text. Some examples are: line 91, please define mPFC; line 104, aPFC.

Line 225: “A different pattern has been observed at synaptic level. In PFC synaptic membranes, no significant changes were observed in the GluN1 subunit (Fig. 7a)”. Please correct “has been” with was/is since the authors are going to present a new result in figure 7.

Correct all sub-indexes in salts and compounds: Example NaH2PO4 à NaH2PO4

Lines 454 and 466 are repeated. “Normal distribution was verified using Kolmogorov–Smirnov test”.

In all figure legends mean + SEM. must be corrected with:  mean ± SEM.  

Line 475 add “male rats” since the experiments were only conducted in this gender.

Figures 2 a and b have only one division (dashed line) and figure 2 c has three. Please unify.

In vivo and in vitro should be in italics.

Author Response

I have reviewed the manuscript “Dopamine-dependent ketamine modulation of glutamatergic 2 synaptic plasticity in the prelimbic cortex of adult rats exposed 3 to acute stress” by Lia Forti and coworkers. The authors study whether acute subanesthetic ketamine administration 6 h after receiving an acute stress (footshock) has any impact on neuronal plasticity (LTP) and expression or activation by phosphorylation of ionotropic glutamate receptors subunits in the pre frontal cortex of male rats.

I find the manuscript appropriate for publication in the International Journal of Molecular Sciences. However, I do have some suggestions and comments that I think will help the authors to improve the clarity for the reader of this article.

We thank the reviewer for appreciating our ms.

I suggest the authors to add more information concerning why they have chosen to test two different DA protocols (lines 127-133) in the results section. Although information is provided in the discussion section of why the authors firstly used 3 uM DA (lines 248-258) providing this information earlier in the manuscript will help the reader to better understand why those two experimental procedures were performed.

We thank the reviewer for the suggestion. We started with a transient perfusion with DA 50 uM in the absence of GABAa block because this protocol was previously shown to facilitate LTP in the PFC of adolescent rats exposed to acute mild stress (Lamanna J et al., J Neurosci Res. 2021;99(2):662-678. doi: 10.1002/jnr.24732). However, this protocol yielded a mild and highly variable potentiation in our hands. Thus, we moved to a different protocol complementing DA application with partial GABAa block (Otani S, Bai J, Blot K. Dopaminergic modulation of synaptic plasticity in rat prefrontal neurons. Neurosci Bull. 2015 Apr;31(2):183-90. doi: 10.1007/s12264-014-1507-3) and obtained a larger and more consistent potentiation. In the revised document, we have specified also in the Results the reasons for the choice of two different DA protocols and inverted panels b and c to show 50 uM DA before the 3 uM protocol (p. 4).

Why have the authors performed their experiments only on male rats? At least a comment for this preference should be stated. Acute traumatic stress can precipitate several psychiatric disorders and gender should be taken into account.

We agree that the comparison of the effects of stress and ketamine in the two sexes is a key issue, which must be addressed in future studies. A comment has been added to the conclusion (p. 14).

Although most abbreviations are defined in the methodology section, it would help the reader to acknowledge them the first time of appearance in the text. Some examples are: line 91, please define mPFC; line 104, aPFC.

As suggested, we have defined abbreviations following the order of appearance in the text.

Line 225: “A different pattern has been observed at synaptic level. In PFC synaptic membranes, no significant changes were observed in the GluN1 subunit (Fig. 7a)”. Please correct “has been” with was/is since the authors are going to present a new result in figure 7.

As suggested, we have corrected the sentence.

Correct all sub-indexes in salts and compounds: Example NaH2PO4 à NaH2PO4

As suggested, we have corrected all sub-indexes in the “Materials and Methods” section.

Lines 454 and 466 are repeated. “Normal distribution was verified using Kolmogorov–Smirnov test”.

As suggested by the referee, we have removed the repetition.

In all figure legends mean + SEM. must be corrected with:  mean ± SEM.  

The errors have been corrected in the figure legends.

Line 475 add “male rats” since the experiments were only conducted in this gender.

The specification of the sex of the animals used in the work has been added in the conclusions (p. 14).

Figures 2 a and b have only one division (dashed line) and figure 2 c has three. Please unify.

The layout of the graphs in Figure 2 has been unified.

In vivo and in vitro should be in italics.

As suggested, we have corrected the font.